# Long-Lasting, Patient-Controlled, Procedure-Free Contraception: A Review of Annovera with a Pharmacist Perspective

**DOI:** 10.3390/pharmacy8030156

**Published:** 2020-08-28

**Authors:** Jennifer J. Virro, Kathleen Besinque, Christiane E. Carney, Danielle Gross, Brian Bernick, Sebastian Mirkin

**Affiliations:** 1TherapeuticsMD, Boca Raton, FL 33431, USA; ccarney@TherapeuticsMD.com (C.E.C.); dgross@tulane.edu (D.G.); bbernick@TherapeuticsMD.com (B.B.); 2School of Pharmacy, Chapman University, Orange, CA 92866, USA; besinque@chapman.edu

**Keywords:** Annovera, contraception, Combined Hormonal Contraception (CHC), vaginal ring, family planning, birth control, hormonal contraception, vaginal system

## Abstract

Annovera (segesterone acetate and ethinyl estradiol vaginal system) is a US Food and Drug Administration FDA-approved long-lasting, reversible contraceptive that is fully administered by the user and does not require a procedure for insertion or removal. The vaginal system is in the shape of a ring and contains low doses of a novel progestin, egesterone acetate, and ethinyl estradiol. It is made of silicone and is fully pliable and flexible. The vaginal system is reusable for 13 cycles, using a 21 days in/7 days out regimen, providing women with the ability to control their fertility. Particularly now during the COVID-19 pandemic when access to contraception has been further reduced, patients may benefit from a method that is both long-lasting and patient-controlled.

## 1. Introduction

Approximately 45% of pregnancies in the United States are unintended [1]. This high rate of unintended pregnancies underscores the need to develop contraceptive methods that meet the needs of women. While over 99% of sexually active women (ages 14 to 44) have used at least one contraceptive method in their lifetime, only 60% of women are actively using contraception [2]. Additionally, many women do not use contraceptive products consistently, with 41% of unintended pregnancies caused by inconsistent use of contraceptives [3]. This can partly be attributed to resupply challenges. In a 2011 study, 38.5% of women using the pill, patch, or ring had two or more late refills during an 18 month period [4]. Short-acting contraceptive methods are typically dispensed in only 1- or 3-month supplies; therefore, women are required to make multiple trips to their clinic or pharmacy for continued access. These additional trips introduce a potential barrier to consistent use, as woman who are late to start their next month’s regimen of contraception will be at an increased risk of pregnancy. To address this issue, the Centers for Disease Control recommend that women be provided with a 1-year supply of birth control [5]. Studies have shown that women who receive a 1-year supply are 30% less likely to have an unintended pregnancy compared to women who receive a 1 to 3 month supply [6,7].

Another strategy to reduce unintended pregnancies is to increase the use of long-acting reversible contraception (LARC) methods such as intrauterine devices (IUDs) and implants. While these methods are highly effective, they must be administered by a healthcare professional and require a procedure for insertion and removal. This can create access issues for some patients since only 56% of office-based OB/GYNs, family practitioners, and adolescent medicine specialists offer on-site IUDs, and only 32% offer implants [8]. Given the limitations associated with these methods, women may benefit from a contraceptive option that is long-lasting, patient-controlled, and procedure-free. 

Yet another strategy to improve patient access to contraception and subsequently reduce unintended pregnancies has been to implement legislation to expand the scope of pharmacists, allowing them to directly prescribe and dispense hormonal contraception without a medical prescription [9]. This strategy removes several access barriers for patients and is a service that is safe and desired by patients. Receiving healthcare directly from a pharmacist does not require an appointment, and many pharmacies have hours that are expanded from typical medical clinic hours, allowing patients to seek services at times that are convenient for them. Additionally, findings from a recent study show that women receiving contraception from pharmacists were younger, had less education, and were more likely to be uninsured than women seeing traditional healthcare providers [10]. It has also been found that patients who receive a 6 month or greater supply of contraception have higher rates of contraceptive continuation and adherence [6,10]. These findings suggest that the pharmacist prescribing of contraception allows for an improved access to birth control methods, especially in underserved populations, and provides a promising strategy to promote consistent use of contraception [11]. Furthermore, patients are receptive to the idea of pharmacists prescribing their birth control method. The results of a national survey of women at risk for unintended pregnancy found that 68% of women liked the idea of being able to obtain contraception directly from a pharmacist, without having to visit a healthcare provider first [12]. Lastly, this is a safe practice. The safety of hormonal contraception is well established, and research has shown that pharmacists are able to safely assess medical contraindications for hormonal contraceptive use and patients are also able to safely self-screen for contraindications to hormonal contraceptive use [13,14].

In addition to typical barriers patients may encounter when attempting to seek contraception, the COVID-19 pandemic has led to sweeping changes across the country, including reduced access to healthcare and the loss of employment and health insurance for millions of Americans. Women have been disproportionately affected by the current health crisis and have been more likely than men to have lost their jobs [15]. According to a recent national survey conducted by the Guttmacher Institute, pandemic-related stress related to finances, job stability, and insecurity about the future has also led to a shift in how women feel about having children. More than 40% of women reported that because of the COVID-19 pandemic, they have changed their plans about when to have children or how many children to have, with 34% wanting to get pregnant later or wanting fewer children because of the pandemic. Thirty-three percent of women reported that they had trouble getting their birth control or had to delay or cancel visiting their healthcare provider for reproductive healthcare. Overall, 23% of women reported thinking more about getting a longer-acting contraceptive method [16]. In light of these findings, a long-lasting, patient-controlled, procedure-free method of contraception is especially relevant in the current climate.

Annovera is a new contraception method that was FDA-approved in 2018 and is available currently in the United States. Annovera is a soft, flexible ring that is self-administered and does not require a procedure. It is inserted into the vagina and remains in place for 21 days and then removed for 7 days. Upon removal, the ring is washed and stored in its case at room temperature; due to the method in which the steroids are dispersed in the ring, it does not require refrigeration at any time [17]. The same ring can be reused for up to a full year or 13 menstrual cycles [17]. As such, the patient can obtain a year’s supply of contraception at the pharmacy that is reversible and fully under the patient’s control. This article will review the clinical data supporting Annovera and provide best practices for patient counseling tailored to pharmacists. 

## 2. Composition of Annovera

Annovera is composed of a silicone elastomer vaginal ring that is opaque white in color (Figure 1). It measures 56 mm in outside diameter and 8.4 mm in cross-sectional diameter. The ring body contains two steroid-releasing channels: an 11 mm core that releases segesterone acetate and an 18 mm core that releases both ethinyl estradiol and segesterone acetate.

Annovera contains a total of 103 mg segesterone acetate and 17.4 mg ethinyl estradiol. The average daily release of segesterone acetate is 150 mcg, and the average daily release of ethinyl estradiol is 13 mcg. At the end of all 13 cycles, 41.3 mg segesterone acetate is released (about 40%) and 3.4 mg ethinyl estradiol is released (19.5%) [17].

## 3. Segesterone Acetate

Segesterone acetate is a novel progestin derived from progesterone. In contrast, many progestins used in combined hormonal contraception are derived from testosterone, and their androgenic activity may be responsible for changes in lipid profile, acne, and weight gain [18,19,20,21]. Segesterone acetate has 100-times higher progestational activity than progesterone as well as high anti-ovulatory potential for the prevention of pregnancy [18,19]. Segesterone acetate lacks androgenic activity at contraceptive doses [18,19]. Segesterone acetate also lacks glucocorticoid activity at contraceptive doses, which may affect water retention [18]. Segesterone acetate is inactive orally and is, therefore, well-suited for administration via long-term sustained release delivery systems [18,19]. Segesterone acetate has been studied in various progestin-only contraceptive systems including vaginal rings, subdermal implants, and transdermal delivery systems for over 20,000 combined cycles [21]. No thromboembolic events were reported in these 20,000 cycles. 

## 4. Clinical Data

### 4.1. Efficacy

The efficacy of Annovera was studied in two 1 year identically designed, multicenter, open-label, single arm, phase 3 trials in the United States and international sites. Two thousand and sixty-five healthy, sexually active, nonpregnant, nonsterilized women aged 18–40 who did not intend to become pregnant over the next 13 months were enrolled. Mean age was 26.7 years; mean BMI was 24.1 kg/m^2^ (range 16.0–41.5); and race was reported as 71.2% Caucasian, 14.1% Black, 3.5% Asian, and 11.2% other/multiple; 28.7% of patients identified their ethnicity as Hispanic [22]. Participants were counseled to follow a 21 days in, 7 days out regimen for up to 13 cycles, and daily paper diaries were used to record vaginal system use [22]. Patients were also counseled not to remove the vaginal system for more than 2 h per cycle.

The Pearl Index for the primary efficacy group was 2.98 (CI: 2.13, 4.06) per 100 woman-years [22]; Annovera’s contraceptive efficacy was comparable to other combined hormonal contraceptives that are under a woman’s control [22]. The Pearl Index was highest among the youngest women (18–19 years) and decreased as age increased. Patients older than 35 years had a Pearl Index of 0.99 (95% CI 0.06–4.34). The Pearl Index was not influenced by BMI [22].

Kaplan–Meier analysis determined that Annovera is 97.5% effective [22]. Furthermore, pregnancy occurrence was similar across all cycles [22].

### 4.2. Patient Acceptability

An acceptability substudy was also performed to determine whether this method of contraception could be considered a successful option in meeting the reproductive needs of patients, as well as identify factors associated with successful use of Annovera. Overall, 89% of patients were satisfied with their use of Annovera [23]. Satisfaction was related to ease of use, side effects, expulsions/feeling the product, and physical effect during sexual activity [23]. After the study, 78% of women indicated that they would prefer to use Annovera as their choice of contraception moving forward [23].

In terms of the effect of Annovera on sexual activity, 69% of women in the clinical trials reported that they never felt Annovera during intercourse [24]. Additionally, 84% of women reported no change or an increase in sexual activity, and 86% of women reported no change or an increase in sexual pleasure with Annovera [24]. Most women, 78%, reported that Annovera never affected their partner’s sexual pleasure [24]. Given this data, patients can be counseled to leave Annovera in place during sexual activity. 

This acceptability substudy also helped to identify several key topics that are important to include in the patient counseling conversation. While most women reported that Annovera was easy to use, problems with ring removal were associated with dissatisfaction, indicating that it is equally important to counsel patients about the appropriate technique for both insertion and removal [23]. This discussion is also important whether the patient has had previous experiences with vaginal rings and/or tampons; there was no association between prior use of vaginal rings or tampons and satisfaction and adherence to Annovera [23].

Anticipatory guidance about the possibility of Annovera expelling and/or feeling Annovera is also important. Patients who felt Annovera while it was in place in the vagina were more likely to be dissatisfied [23]. As such, it is important to counsel patients that Annovera should be positioned in the upper two-thirds of the vagina and that they should not feel it after insertion. Counseling about the possibility of expulsions, especially during voiding, undertaking activities that require straining, or performing certain daily activities, such as squatting, should also be included in the counseling conversation [23].

### 4.3. Bleeding Profile 

Bleeding patterns associated with Annovera were well tolerated. When the vaginal system was removed cyclically as directed, 98% of women had predictable, scheduled bleeding [25]. Women experienced an average of 3.3 bleeding days per cycle [25]. Unscheduled bleeding and spotting occurred in 5–10% of patients and lasted for an average of 1 day or less per cycle [17]. Approximately 3–5% of patients had amenorrhea each cycle [25]. Only 1.7% of women discontinued Annovera due to unacceptable bleeding [26]. 

### 4.4. Vaginal Microbiome

Data from a microbiology substudy showed that the use of Annovera did not increase the rate of vaginal infection and was not disruptive to the vaginal ecosystem [27]. Gynecological examinations were conducted, and vaginal swabs were obtained for wet mount microscopy, Gram stain, and culture. At the end of the 13-cycle study period, Annovera was removed from the vagina and cultured. There were no significant changes in the detection rate of bacterial vaginosis (BV), vulvovaginal candidiasis (VVC), or trichomoniasis throughout the study period [27]. Additionally, there was a high level of agreement between the vaginal cultures and the cultures obtained from the surface of Annovera [27]. The results suggest that the ring surface does not promote the proliferation of microorganisms, most notably, *Staphlococcus aureus*, which has been suggested as a causative factor of toxic shock syndrome [27].

### 4.5. Adverse Reactions 

The safety of Annovera was demonstrated in three 13-cycle, open-label clinical trials that enrolled 2308 healthy females aged 18–40 years across 20 United States sites and 7 international sites [26]. These women contributed 21,590 cycles of exposure for safety evaluation [26]. The most common adverse reactions reported in ≥5% of clinical trial subjects are listed in Table 1 [26]. Most reactions were considered mild, and most occurred within the first cycle, especially during the first few days of initial insertion of the ring. Throughout the clinical trials, 12% of women discontinued Annovera due to an adverse reaction (Table 2) [26]. Additionally, 25% of subjects reported at least 1 complete expulsion during their use of Annovera; 1.4% of women discontinued Annovera due to expulsion of the vaginal ring [26]. When using Annovera, women did not experience a clinically relevant increase in weight [26]. In an acceptability substudy, women experiencing hormonal side effects reported lower levels of satisfaction. It is, therefore, important to counsel patients appropriately about the possibility of experiencing hormonal or other side effects, as well as give advice about managing these experiences [23].

### 4.6. Venous Thromboembolism

Four nonfatal venous thromboembolisms (VTEs) occurred in the clinical trials and all 4 women recovered. Three of the cases occurred in women with risk factors for VTE: one woman had Factor V Leiden that was diagnosed after the event and two women were overweight. The fourth case occurred in a woman who withdrew from the study before a clotting evaluation could be completed. Post marketing studies are being conducted to further evaluate the risks of VTE [28]. Like other combined hormonal contraceptives (CHCs), Annovera contains a boxed warning for women older than 35 years who smoke cigarettes since these women are at an increased risk of cardiovascular events [17]. 

### 4.7. Use in Patients with BMI > 29 kg/m^2^


After approximately 50% enrollment, women with a BMI > 29 kg/m^2^ were excluded from the clinical trials due to the occurrence of 2 VTEs in this population (see Section 4.4). Therefore, out of 209 women with a BMI > 29 kg/m^2^ who enrolled in the study, only 36 completed the trial thereby limiting the amount of clinical data that was collected in this population [26]. Accordingly, while not contraindicated, the product labeling for Annovera has a limitation of use in women with a BMI > 29 kg/m^2^ reflecting the limited population with a BMI > 29 kg/m^2^ that was evaluated in the clinical trials [17]. Annovera’s Pearl Index does not change depending on a woman’s BMI [22]. 

### 4.8. Return to Fertility 

After completion of the Annovera clinical trial, 290 women who were trying to become pregnant or not continuing use of hormonal contraception were followed up for 6 months to assess the return to fertility. One hundred percent of women in this follow-up study reported either the return of normal menses or pregnancy. Of the women who desired pregnancy, 63% became pregnant within the first 6 months [22].

### 4.9. Concurrent Use of Vaginal Products 

Annovera does not protect from sexually transmitted infections. Annovera is compatible with male condoms made with natural rubber latex, polyisoprene, and polyurethane [17]. If there is a need to treat a vaginal condition, water-based vaginal creams or oral therapies should be used rather than oil-based vaginal suppositories [17]. Co-administration of oil-based miconazole vaginal suppositories with Annovera was associated with increased systemic exposure of segesterone acetate and ethinyl estradiol, whereas co-administration with a water-based cream miconazole formula was not [29]. Similarly, if women want to use a vaginal lubricant, only water-based lubricants should be used. Silicone and oil-based lubricants, including coconut oil, will alter the Annovera vaginal system and should be avoided. 

## 5. Pharmacist Viewpoint

Annovera is a unique contraceptive, offering women an option for a self-administered method of contraception that is long-lasting and easy to use. As a relatively new product, women may not be familiar with the system. Pharmacists can play an important role in answering questions and highlighting some of the important differences between Annovera and other hormonal contraceptives. In states where pharmacists prescribe self-administered hormonal contraceptives, Annovera may be an option to present to women interested in a long-lasting method that does not require daily pill taking. It may be important to mention to potential users that the contraceptive ring systems are not interchangeable, and instructions for use are specific to each product. 

One of the key features of the Annovera system is its convenience. Upon initiation, the system should be inserted vaginally for 21 days, removed for 7 days and then reinserted in this fashion for 13 cycles, providing a year of contraceptive protection. Because the same ring system is used for a year, women do not have to return to the pharmacy or worry about having enough of their contraceptive with them if they travel or are away at school. There is no need for a daily reminder to take a pill or weekly reminder to change a patch and having a full year of contraception without having to worry about obtaining supplies for a year is convenient. The system should be stored at room temperature; no refrigeration is needed. 

In providing patient consultation, pharmacists can inform users that Annovera is a well-tolerated method of long-lasting, reversible contraception. Reported side effects are generally mild, self-limiting, and are most often reported within the first cycle of use. Users can expect side effects similar to those of other hormonal contraceptives such as headache, mild nausea, and some menstrual changes during the initial use period. The system is flexible and can be inserted and removed using clean fingers. Once inserted, the system should remain in place and is not noticeable to most users. At the end of the 13 cycles, a new system can be prescribed for another year of contraception. A small compact case is provided with the product for use during system removal days for discrete storage. 

Women receiving Annovera may have questions about using the product. Frequently asked questions/counseling points include: 

### 5.1. Prior to Using Annovera 

The ring system should be washed with mild soap and water then patted dry with a towel before being inserted into the vagina. The system should be inserted as high into the vagina as is comfortable. 

### 5.2. How to Start 

Women who are initiating Annovera begin using the system as with other hormonal contraceptives. Potential users should carefully read and understand the patient prescribing information [18]. For patients who have no hormonal contraceptive use in the preceding cycle or after Copper IUD removal, the recommended regimen is for the patient to insert Annovera between ays 2 and 5 of her regular menstrual period; no back-up contraception is needed. If menstrual cycles are irregular or if the start is more than 5 days from the last menstrual bleeding, the woman should use an additional barrier method during coitus, such as a male condom or spermicide, for the first 7 days of Annovera use. Women already using a combination hormonal contraceptive method consistently and correctly and who you are reasonably certain is not already pregnant may switch from her previous CHC to Annovera on any day of the CHC cycle (day 1–28), without the need for back-up contraception, but no more than 7 hormone-free days should occur before starting Annovera. No back-up is required for women switching CHCs [17]. If a patient has no contraindications to the use of ethinyl estradiol (EE), she may elect to switch from a progestin-only method to Annovera. If switching from progestin-only pills (POP), she should begin Annovera at the time she would have taken her next POP pill. If switching from an injection, she should begin Annovera at the time of her next scheduled injection. If switching from an implant or intrauterine system (IUS), she should begin Annovera at the time of implant or IUS removal. In all these cases, the woman should use a back-up method for the first 7 days of Annovera use [17]. If a woman has no contraindications to the use of EE, Annovera may be initiated for contraception within the first 5 days following a complete first trimester abortion or miscarriage without additional back-up contraception. If more than 5 days have elapsed from the first trimester abortion or miscarriage, then follow the instructions for patients who have not had hormonal contraceptive use in the preceding cycle and a barrier method should be used from the time of the first trimester abortion or miscarriage to the initiation of Annovera. Annovera should not be started earlier than 4 weeks after a second trimester abortion or miscarriage due to the increased risk of thromboembolism [17]. Annovera should not be started sooner than 4 weeks postpartum and only in females who choose not to breastfeed. Prior to 4 weeks postpartum, there is an increased risk of thromboembolism. The initiation of Annovera 4 weeks or more postpartum should be accompanied by an additional back-up method for the first 7 days if the woman has not yet had a period. Consider the possibility of ovulation and contraception occurring prior to initiating Annovera. Females who are breastfeeding should not use Annovera until after weaning [17].

### 5.3. What Is the “Cycle” for Use

The ring system once inserted on day 1 should remain in place for 21 consecutive days. The ring system can be removed for limited periods of time if desired but not exceeding 2 h over the course of the 21 days [17]. Although the system can be removed for intercourse, it is not necessary. On day 21, the ring system is removed and should be washed, dried, and stored in the compact case provided. The ring system should be stored in the case when not in use. 

### 5.4. What If the System Comes Out 

If the system comes out, it can be cleaned and reinserted. As long as the system has not been out of the vagina for more than 2 h over the cycle, no backup method is needed [17]. If the ring has been out of the vagina for more than 2 h during the cycle for any reason, a backup method of contraception should be used for the next 7 days [17]. 

### 5.5. What to Do If the System Is Inserted Late or Early for Subsequent Cycles 

If the system is reinserted after being out only 5–6 days, the system can remain in place until the normal removal day [17]. If the system is inserted late, after being out more than 7 days, the system should remain in place for the next 21 days and day 1 adjusted for future cycles [17]. A backup method of protection should be used for the first 7 days. If the system is removed late, after more than 21 days, the system should be removed as soon as remembered and reinserted 7 days later [17]. If the system is removed 1–2 days early, the system should be reinserted 7 days later. The system can be left in place during that next cycle until the usual removal date (22–23 days) or removed after 21 days to set a new removal day [17]. 

### 5.6. How to Dispose of the System 

The ring system should be disposed of using a pharmaceutical “take-back” bin or returned to pharmacies with medication disposal services when possible. Users can also search the internet to obtain an envelope for mailing the system to a pharmaceutical waste disposal company. If a take-back option is unavailable, then discard in the waste receptacle out of reach of children and pets. The system should not be flushed or cut-up and disposed of in the garbage [17]. 

## 6. Market Access and Coverage

As of June 2020, Annovera has 66% nationwide unrestricted access for patients with commercial insurance. For Medicaid patients, Annovera has 46% unrestricted access in 38 states and Washington, DC. States where Medicaid coverage exists are: AK, AL, AR, CO, CT, DC, FL, HI, ID, Il, IN, KS, KY, LA, MA, MD, ME, MI, MO, MT, NC, ND, NE, NH, NJ, NM, NV, NY, OH, OR, RI, SC, SD, TN, TX, UT, VA, WA, and WY. For more information, please reach out to TherapeuticsMD. 

## 7. Conclusions

Annovera is a new, long-lasting, reversible contraceptive that is procedure-free and allows women to be in control of their fertility for 1 year. Pharmacists nationwide may be dispensing this method and counseling patients about its use. In states where pharmacists have contraception prescribing authority, Annovera is the only long-lasting method that a pharmacist can prescribe and dispense on site. The efficacy and safety profile of Annovera is similar to other CHCs, but it has the advantage of supplying the patient with a 1 year supply of birth control after only one trip to the pharmacy. Increased patient interest in long-acting methods, coupled with the unique attributes of Annovera, makes this an important and relevant contraceptive option for patients, especially amidst the current health crisis. 

## Figures and Tables

**Figure 1 pharmacy-08-00156-f001:**
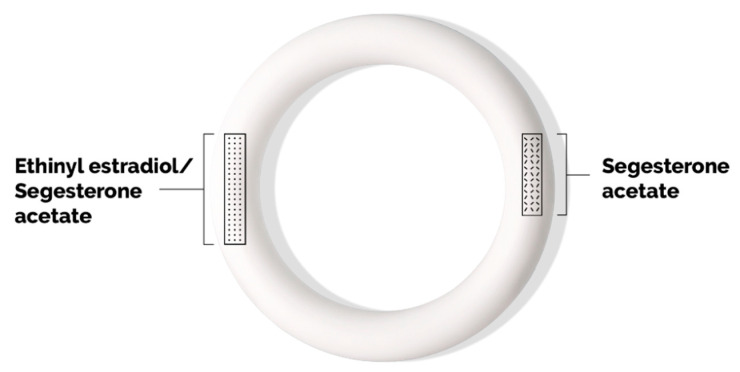
Annovera vaginal system with hormone cores containing segesterone acetate and ethinyl estradiol.

**Table 1 pharmacy-08-00156-t001:** List of most common adverse reactions reported in ≥5% of Annovera clinical trial subjects.

Adverse Reactions Reported by ≥5% of Subjects	% (*n* = 2308)
Headache, including migraine	38.6
Nausea/vomiting	25.0
Vulvovaginal mycotic infection/vaginal candidiasis	14.5
Abdominal pain/lower/upper	13.3
Dysmenorrhea	12.5
Vaginal discharge	11.8
UTI/cystitis/pyelonephritis/genitourinary tract infection	10.0
Breast pain/tenderness/discomfort	9.5
Metrorrhagia/menstrual disorder	7.5
Diarrhea	7.2
Genital pruritus	5.5

**Table 2 pharmacy-08-00156-t002:** Adverse reactions leading to discontinuations in ≥1% of Annovera clinical trial subjects.

Adverse Reactions Leading to Discontinuation by ≥1% of Subjects	% (*n* = 2308)
Metrorrhagia/menorrhagia	1.7
Headache, including migraine	1.3
Vaginal discharge/vulvovaginal mycotic infections	1.3
Nausea/vomiting	1.2

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
