# Peer review of "Long-Lasting, Patient-Controlled, Procedure-Free Contraception: A Review of Annovera with a Pharmacist Perspective"

_pharmacy, 2020, doi:10.3390/pharmacy8030156_

Round 1

Reviewer 1 Report

One minor suggestion I have is for patients who newly start Annovera and are not switching from another CHC, include information about back-up contraception for the first 7 days.  That point isn't completely clear in the manuscript.  

The Annovera package insert also states the product may be discarded via a trash receptacle, out of the reach of children and pets, if a medication take-back option is unavailable.  The manuscript specifically states not to place in the trash.  While a medication take-back receptacle is always the best option, it may be difficult for a young, college-age woman using Annovera to find a take-back option.  Consider rewording to make consistent with the PI.

Author Response

One minor suggestion I have is for patients who newly start Annovera and are not switching from another CHC, include information about back-up contraception for the first 7 days.  That point isn't completely clear in the manuscript.  

Thank you – section 5.2 has been updated to include more information about how to start Annovera in a wider range of patients (information taken directly from PI).

The Annovera package insert also states the product may be discarded via a trash receptacle, out of the reach of children and pets, if a medication take-back option is unavailable.  The manuscript specifically states not to place in the trash.  While a medication take-back receptacle is always the best option, it may be difficult for a young, college-age woman using Annovera to find a take-back option.  Consider rewording to make consistent with the PI.

Updated (section 5.6).

Reviewer 2 Report

In general provides interesting and useful information on Annovera.  Recommend authors read through with careful eye towards grammar, writing style, and active (vs passive) voice.  Some specific recommendations are as follows:

  • the concept of a annual contraceptive is mentioned in the context of COVID-19.  While the access issues noted are true, these issues would be true during any occasion of reduced access (travel, school, etc).  Recommend broadening this discussion a bit. 
  • pg 1, ln 66 - recommend omitting date in this sentence so readers are not discouraged by decade-old data
  • pg 1, reference #2: more recent data here: https://www.cdc.gov/nchs/data/databriefs/db173.pdf
  • pg 2, ln 99: reference seems out of place (11), and probably should be cited in earlier paragraph where this issue is addressed. I don't believe many pharmacists, if any, were providing HC in 2011. 
  • pg 2, ln 109: both of these references address patient self-screening; also - healthcare providers besides physicians have been proving HC so it seems odd to call out pharmacists as "non-physicians here"
  • pg 2, ln 123 (similar to comment above): and how might women benefit in other similar situations with reduced access?
  • pg 2, ln 137: mention of refrigeration repetitive with line 128 above
  • pg 3, ln 145: good content in this paragraph but it could be re-organized for better flow; particularly the comparison of segesterone with other progestins 
  • pg 3, ln 152-153: This sentence does not make sense as written...a compound's oral activity or inactivity has nothing to do with how suited it is for a sustained release delivery system. For example, levonorgestrel is in long-term sustained release delivery systems (IUDS) but can also be given orally. If there are pharmacokinetic data that would better support the use of segesterone acetate in this delivery system please briefly include them here. 
  • pg 4, ln 252: annovera has an initial expulsion rate of about 25%. This information should be included along with what to tell women when it happens. Authors can determine if it is better here or in counseling points.
  • pg 6, ln 302: Are post-marketing studies being conducted in women with elevated BMI as well? If so - please mention. 
  • pg 6, ln 308: how does this compare to women who have not used HC? (ie - is 63% what we should expect)
  • pg 6, ln 318: This section has a lot of great information but leaves me with a lot of questions as well: does the increased systemic exposure to segesterone and EE have any clinical impact? How long do the drug levels stay elevated? How do silicone and oil-based products "alter the Annovera vaginal system"? 
  • pg 7, ln 362: does it have to be removed for 7 days each cycle? or can the system be left in for a longer period, such as 84 days?

Author Response

  • pg 1, ln 66 - recommend omitting date in this sentence so readers are not discouraged by decade-old data

        Date removed

  • pg 1, reference #2: more recent data here: https://www.cdc.gov/nchs/data/databriefs/db173.pdf

        Reference updated

  • pg 2, ln 99: reference seems out of place (11), and probably should be cited in earlier paragraph where this issue is addressed. I don't believe many pharmacists, if any, were providing HC in 2011. 

        Verbiage changed in text to reflect the reference more accurately

  • pg 2, ln 109: both of these references address patient self-screening; also - healthcare providers besides physicians have been proving HC so it seems odd to call out pharmacists as "non-physicians here"

        Updated

  • pg 2, ln 123 (similar to comment above): and how might women benefit in other similar situations with reduced access?

        Paragraph 67-80 discusses access in light of COVID

  • pg 2, ln 137: mention of refrigeration repetitive with line 128 above

        Deleted

  • pg 3, ln 145: good content in this paragraph but it could be re-organized for better flow; particularly the comparison of segesterone with other progestins 

        Reorganized

  • pg 3, ln 152-153: This sentence does not make sense as written...a compound's oral activity or inactivity has nothing to do with how suited it is for a sustained release delivery system. For example, levonorgestrel is in long-term sustained release delivery systems (IUDS) but can also be given orally. If there are pharmacokinetic data that would better support the use of segesterone acetate in this delivery system please briefly include them here. 

        Additional reference added

  • pg 4, ln 252: annovera has an initial expulsion rate of about 25%. This information should be included along with what to tell women when it happens. Authors can determine if it is better here or in counseling points.

        Added, line 186-187

  • pg 6, ln 302: Are post-marketing studies being conducted in women with elevated BMI as well? If so - please mention. 

        Yes. This is referred to in section 4.6: “Post marketing studies are                      being conducted to further evaluate the risks of VTE [29].”

  • pg 6, ln 308: how does this compare to women who have not used HC? (ie - is 63% what we should expect)

        We do not have data on this. Different products define return to fertility              different. In the clinical trial for Annovera return to fertility was defined as          return of normal menses or pregnancy within a 6-month period.

  • pg 6, ln 318: This section has a lot of great information but leaves me with a lot of questions as well: does the increased systemic exposure to segesterone and EE have any clinical impact? How long do the drug levels stay elevated? How do silicone and oil-based products "alter the Annovera vaginal system"? 

        The content in this paragraph is what we have been approved to say.

  • pg 7, ln 362: does it have to be removed for 7 days each cycle? or can the system be left in for a longer period, such as 84 days?

        This is currently an off-label use and we are not approved to comment on          this. The clinical trial data was obtained using a 21/7 regimen and we                have no data on any other regimens

This manuscript is a resubmission of an earlier submission. The following is a list of the peer review reports and author responses from that submission.